# The Relative Abundance and Occurrence of Sharks off Ocean Beaches of New South Wales, Australia

**DOI:** 10.3390/biology11101456

**Published:** 2022-10-04

**Authors:** Kim I. P. Monteforte, Paul A. Butcher, Stephen G. Morris, Brendan P. Kelaher

**Affiliations:** 1National Marine Science Centre, Southern Cross University, Coffs Harbour, NSW 2450, Australia; 2NSW Department of Primary Industries, National Marine Science Centre, Coffs Harbour, NSW 2450, Australia; 3NSW Department of Primary Industries, Wollongbar, NSW 2477, Australia

**Keywords:** *Carcharodon carcharias*, *Carcharhinus*, *Carcharias taurus*, nearshore, coastal, shark, drone

## Abstract

**Simple Summary:**

Coastal sharks are especially susceptible to anthropogenic pressures due to their use of nearshore habitats. It is therefore necessary to distinguish species-specific variation in occurrence among ocean beaches and identify key environmental drivers that may influence shark distribution patterns. For swell-influenced coastal beaches, however, information on shark abundance and occurrence is still limited. The relative abundance and occurrence of sharks off 42 ocean beaches of New South Wales (NSW), Australia, was investigated using a long-term drone-based monitoring program from 2017–2021. Overall, there were 36,384 drone flights conducted, sighting a total of 281 sharks. Shark observations occurred in <1% of drone flights, indicating that potentially dangerous sharks, and sharks in general, are rare off NSW beaches. Key environmental predictors, such as the distance to the nearest estuary, headland, and island, as well as ocean temperature and wave height, were identified for species-specific shark distribution. This study demonstrated that existing drone-based monitoring programs used to reduce shark bite incidents can also provide valuable information about the distribution of potentially dangerous, vulnerable, and endangered coastal sharks. This information will be beneficial for implementing evidence-based conservation strategies and may also assist in minimising negative human-shark interactions.

**Abstract:**

There is still limited information about the diversity, distribution, and abundance of sharks in and around the surf zones of ocean beaches. We used long-term and large-scale drone surveying techniques to test hypotheses about the relative abundance and occurrence of sharks off ocean beaches of New South Wales, Australia. We quantified sharks in 36,384 drone flights across 42 ocean beaches from 2017 to 2021. Overall, there were 347 chondrichthyans recorded, comprising 281 (81.0%) sharks, with observations occurring in <1% of flights. Whaler sharks (*Carcharhinus* spp.) had the highest number of observations (*n* = 158) recorded. There were 34 individuals observed for both white sharks (*Carcharodon carcharias*) and critically endangered greynurse sharks (*Carcharias taurus*). Bull sharks (*Carcharhinus leucas*), leopard sharks (*Stegostoma tigrinum*) and hammerhead species (*Sphyrna* spp.) recorded 29, eight and three individuals, respectively. Generalised additive models were used to identify environmental drivers for detection probability of white, bull, greynurse, and whaler sharks. Distances to the nearest estuary, headland, and island, as well as water temperature and wave height, were significant predictors of shark occurrence; however, this varied among species. Overall, we provide valuable information for evidence-based species-specific conservation and management strategies for coastal sharks.

## 1. Introduction

As apex predators and secondary consumers, sharks play an important role in maintaining the structure and function of marine ecosystems [1,2,3]. The loss of sharks from marine habitats can, therefore, negatively affect ecosystem dynamics and resilience [4,5,6]. Global shark abundances are declining due to habitat loss, ocean exploitation, unsustainable fishing practices, pollution, climate change and lethal methods of shark mitigation [7,8,9]. Coastal sharks are, however, especially susceptible to anthropogenic pressures due to their use of nearshore habitats [10,11].

Coastal sharks utilise a variety of highly productive and nutrient-rich nearshore habitats for foraging and reproduction sites [12,13]. The varied use of these nearshore environments among shark species is driven by biological characteristics and ecological processes, such as prey abundance and resource availability, as well as the presence of other competitors or predators [14,15,16]. Additionally, shark distribution within coastal habitats is influenced by environmental factors such as water temperature and salinity [11,17,18], as well as the proximity to productive habitats, such as headlands [19], reefs [20,21,22] and estuaries [23,24].

Despite concerns around declining shark populations, conservation and management of many shark species is challenging due to their migratory nature and their wide-ranging distributions [25,26]. Identifying species-specific variation in occurrence among ocean beaches will be beneficial for implementing evidence-based conservation strategies. Additionally, conservation efforts for threatened shark populations can be improved by establishing the potential environmental drivers that influence habitat use for different coastal species [17,27]. Investigating shark occurrence off ocean beaches may also assist in minimising negative human-shark interactions by identifying the movement patterns of potentially dangerous species [28,29]. For swell-influenced coastal beaches, however, information on shark abundance is limited (but see Colefax et al. [30] and Kelaher et al. [31]).

We investigated the relative abundance and occurrence of sharks off ocean beaches of New South Wales (NSW), Australia, using a long-term drone-based monitoring program over 1000 km of coastline or 8.4° of latitude. We based our methods on recent drone-based surveying techniques that successfully quantified sharks [32,33] and other marine wildlife [34] off exposed ocean beaches. Using these techniques, we tested the hypotheses that (i) the relative abundance and occurrence of sharks varies among beaches, and (ii) shark abundance is predictable by environmental factors, such as ocean temperature, wave height and distance to the nearest estuary, headland, and island. Our findings can contribute to conservation strategies for threatened shark species and assist with shark management by detailing shark occurrence off popular ocean beaches. Such information can also directly contribute to optimising shark mitigation efforts to maximise their cost-effectiveness in a globally recognised hotspot for shark bites [28,35].

## 2. Materials and Methods

### 2.1. Study Locations

Data collected from drone surveys were used to investigate the relative abundance and occurrence of coastal sharks off 42 ocean beaches between Kingscliff (−28.255, 153.579) and Pambula, NSW, Australia (−36.941, 149.910; Figure 1).

### 2.2. Survey Methods

Drone flights occurred during the austral spring, summer, and autumn months in 2017/2018 (sampling season 1), 2018/2019 (sampling season 2), 2019/2020 (sampling season 3), and 2020/2021 (sampling season 4). Over the four sampling periods, there was some variation among locations sampled, timing, and sampling frequency (Appendix A, Appendix A). However, the annual flight periods began in September and ended in May of the following year.

Contractors and NSW Surf Life Saving (NSW SLS) drone pilots completed surveys using DJI Phantom 4 and DJI Mavic 2 enterprise quadcopter drones (<2 kg). Drone schedules typically included two flights every hour from 0900 to 1600, with a maximum flight duration of 25 min. Drone pilots undertook all flights during rain-free periods with light to moderate winds to ensure safe operations [30,32].

Surveys were up to 2.0 km in length, monitoring 0.5 to 1.0 km on either side of the pilot located at the control station. Flights occurred directly behind the back of the surf break, and consequently, distances from the shoreline varied depending on swell and tidal conditions. Pilots flew drones at 60 m altitude; however, the drones were often lowered when a shark was present to improve species identification [36].

### 2.3. Video Analysis of Drone Footage

Video footage of all shark observations was collected for post-flight analysis to ensure higher accuracy in species identification [34]. Sharks were identified down to the lowest possible taxonomic level. For most observations, this was to either species (e.g., white sharks, *Carcharodon carcharias*) or genus level (e.g., whaler sharks, *Carcharhinus* spp.). In cases where the type of shark could not be identified with certainty (due to water depth or clarity), it was categorised as an ‘unknown shark’. Additionally, guitarfish species were included in this study as they are commonly misidentified as white sharks during drone surveying, resulting in unnecessary beach evacuations [31]. The beach location, time, and date of all shark and guitarfish observations were also recorded.

### 2.4. Environmental Factors

The sea state (ranging between 1 and 3 on the Beaufort scale), water clarity (scaled between 1 and 5, where 1 is rated as very poor and 5 is rated as very good), and the presence of other marine wildlife, such as fish schools, turtles, and rays (categorised as either present or absent) were recorded for each flight where an observation occurred. Additionally, water temperature and wave height hourly recordings were calculated into daily averages for each flight day at each beach location. Water temperature data were supplied by New South Wales Department of Primary Industries (NSW DPI) Shark-Management-Alert-in-Real-Time (SMART) drumlines and shark listening stations. Wave height data were provided by Manly Hydraulics Laboratory (MHL) and NSW Department of Planning, Industry, and Environment (DPIE).

Using the Path Ruler tool in Google Earth PRO version 7.3.4.8248, the distance from each beach to the nearest permanent open sea estuary, headland, island, and rocky reef was measured. These measurements were taken directly out from the Surf Lifesaving Club of each beach location or the beach car park for locations without a Surf Lifesaving Club. Measurements started 100 m from the shoreline and measured the most direct swim path.

### 2.5. Statistical Analyses

The series of observations for each shark genus at each beach was aggregated to give the number of days when the genus was sighted and the number of days when flights were undertaken. The ratio of days with detections to flight days was then taken as an estimate of detection probability at each beach.

The low detection rates and clustering of beaches within the covariate space precluded multi-factor models of detection probability. However, a ranking and visualisation of the association between detection probability and each covariate was conducted as follows. The relationship between detection probability of each genus and distance to the nearest estuary, headland, and island, as well as environmental variables (mean water temperature and mean wave height), was described by a series of generalised additive models (see Wood [37] for example) with form:$$\log\left(\frac{p_{i}}{1-p_{i}}\right)=s\left(x_{i}\right)+error$$
where *p_i_* is the detection probability and *x_i_* is the covariate value at beach (*i* = 1:42) and *s*() represents a cubic spline basis.

Distance to the nearest rocky reef was not included in analyses due to the high positive correlation with distance to the nearest headland. Additionally, analyses for leopard sharks and hammerhead species were not conducted due to the rarity of their sightings.

A test for the contribution of the spline was conducted as per Wood [37] and tabulated for each genus and covariate. Plots of observed genera-specific detection rates and estimated response curves with a 95% confidence region were then constructed as an aid to data visualisation. Akaikes Information Criteria (AIC) is also presented for each model to show the relative importance of each factor. The model with the lowest AIC for each shark genus is an indicator of the best predictor. The data analyses were conducted in the R environment [38] with particular use of the *mgcv* package [37].

## 3. Results

Pilots completed 36,384 individual drone flights totalling 10,062 h across 4298 flight days (Appendix A). A flight day consisted of 8 or 9 flights (mean 8.5 flights), with an average flight duration of 16.6 min.

A total of 347 chondrichthyans, comprising 281 (81.0%) sharks and 66 (19.0%) rays over 150 days, were recorded. These observations occurred in 3.5% of flight days and in <1% of drone flights. Whaler sharks had the highest observations with 158 individuals, comprising 45.5% of total sightings. White sharks and greynurse sharks both recorded 34 (9.8%) individuals in total, and bull sharks represented 8.4% (*n* = 29) of sightings. There were eight (2.3%) leopard sharks and only three hammerhead sharks (0.9%) sighted. Guitarfish and shovelnose rays represented 19.0% (*n* = 66) of total observations. Unknown sharks comprised 4.3% (*n* = 15) of sightings, and no tiger sharks (*Galeocerdo cuvier*) were recorded.

Evans Head (Main Beach) recorded the highest observations, with 99 individuals sighted, representing 28.5% of total observations. Forster followed with 63 (18.2%) sightings, 26 of which were greynurse sharks, representing 76.5% of total greynurse observations, and had the highest proportion of days where greynurse sharks were observed in relation to total flight days (0.1). Crescent Head (*n* = 5), Port Macquarie (Lighthouse Beach; *n* = 1), Anna Bay (*n* = 1), and Avoca (*n* = 1) were the only other locations to record observations of the critically endangered greynurse species.

Forster also had the highest white shark observations with ten individuals, representing 29.4% of total white shark sightings. Crescent Head, Anna Bay and Pambula locations recorded the second highest abundance of white shark observations (*n* = 4). The highest proportion of days where white sharks were sighted relative to total flight days, however, was at Hawks Nest (0.07), followed by Pambula (0.06). Ballina (Lighthouse Beach and Sharpes Beach) and Anna Bay recorded the highest bull shark observations with five individuals each. However, Anna Bay (0.03) and Scotts Head (0.03) had the highest proportion of bull shark observation days to total flight days. Forster had the highest observations of sharks per flight effort overall, calculated to an average of 1.4 sightings per flight day, followed by Pambula, Hawks Nest and Crescent Head with 0.8 observations per flight day (Appendix A).

Observations of leopard sharks and hammerhead species only occurred in northern NSW locations. Leopard sharks were recorded at Kingscliff (Kingscliff Beach; *n* = 4), Cabarita (*n* = 1), and Byron Bay (Main Beach; *n* = 3), and hammerhead species were recorded at Kingscliff (South Beach; *n* = 2) and Ballina (Shelly Beach; *n* = 1). Most sharks were observed as solitary; however, large aggregations of whaler sharks were seen on three separate occasions at Evans Head (Main Beach; *n* = 64 and 27) and Byron Bay (Main Beach; *n* = 11).

All shark observations occurred in water temperatures ranging from 16.7–26.2 °C, with an average of 22.2 °C, and in wave heights ranging from 0.7–2.5 m, with an average of 1.4 m. There was other marine wildlife present (e.g., fish schools, turtles, rays) on 45.9% of days where shark observations occurred. There were no shark observations recorded at 19 beaches.

### Factors Affecting Species-Specific Shark Distribution

White (*p* < 0.05; Table 1; Figure 2a) and greynurse (*p* < 0.01; Figure 2c) observations were significantly associated with distance to the nearest estuary, indicating that the probability of sighting a white or greynurse shark increased with decreasing distance to the nearest estuary. Analyses of bull sharks and whaler species, however, revealed that distance to the nearest estuary did not explain significant variation in sightings (Figure 2b,d).

Greynurse sharks were significantly associated with distance to the nearest headland (*p* < 0.001; Table 1), with all locations that recorded greynurse sharks being within proximity to headlands (Figure 3c). In comparison, this factor did not explain significant variation for white, bull and whaler shark sightings (Figure 3a,b,d). Distance to the nearest island was the best predictor for whaler shark occurrence (Table 1), and bull (*p* < 0.05), greynurse (*p* < 0.01) and whaler (*p* < 0.001) sharks were significantly associated with this factor. However, for white sharks, distance to the nearest island did not explain significant variation in observations (Figure 4a). Overall, we found that bull, greynurse and whaler shark sightings all decreased significantly with an increase in distance to the nearest island (Figure 4b–d).

Water temperature was the best predictor to explain white and greynurse shark occurrence (Table 1). White (*p* < 0.001), bull (*p* < 0.05), greynurse (*p* < 0.05) and whaler (*p* < 0.01) shark observations were significantly associated with water temperature (Table 1). At locations where white shark observations did occur, these sightings were within water temperatures between 16.7 and 25.7 °C, with an increase in white sharks in temperatures of ~19.5 and ~21.5 °C (Figure 5a). Bull and whaler shark observations occurred within similar water temperature ranges of 17.8 to 25.6 °C and 17.8 to 26.2 °C, respectively (Figure 5b,d). Furthermore, there was an increase in bull and whaler observations in water temperatures between 21.0 and 22.5 °C. Greynurse observations occurred within water temperatures ranging between 20.5 and 22.0 °C (Figure 5c).

Wave height was the best predictor for bull shark occurrence (Table 1), and white (*p* < 0.05), bull (*p* < 0.05), greynurse (*p* < 0.001) and whaler (*p* < 0.05) observations were all significantly associated with wave height (Table 1). Sharks were observed within similar wave height ranges of 0.7 to 2.5 m, with slight variation among species (Figure 6). However, most observation days for white, bull, greynurse and whaler species occurred within wave heights of 1.4 and 1.6 m.

Sharks were observed within all water clarity classifications (Appendix A); however, most shark observations (*n* = 118) occurred on days with ‘poor’ water clarity. There were 58, 44 and 34 sharks observed within ‘good’, ‘OK’ and ‘very poor’ water clarity, respectively. The lowest observations (*n* = 27) occurred on days with a water clarity rating of ‘very good’. Shark observations occurred within a sea state ranging from 1 to 3 (Appendix A), with most observations (*n* = 191) recorded on days with a sea state of 2 and the least observations (*n* = 19) occurring on days with a sea state of 3.

## 4. Discussion

Using the largest drone-based shark monitoring program in the world, we successfully investigated the relative abundance and occurrence of coastal sharks off ocean beaches in south-eastern Australia. We not only documented the species-specific distribution patterns of sharks, but we also highlighted key environmental predictors that help explain these patterns. In particular, we showed that water temperature, wave height, and distances to the nearest estuary, headland, and island were significant predictors of shark occurrence; however, the relative importance of these predictor variables varied among species. Overwhelmingly, our results demonstrated that sharks are rare off the ocean beaches of NSW, with observations occurring in <1% of drone flights.

### 4.1. Shark Abundance and Occurrence

Whaler sharks were the most abundant genera observed and were the only sharks to occur in large aggregations. With the exception of these occurrences, sharks were typically observed as solitary animals. The large whaler shark groups were sighted at Main Beach, Evans Head. This location is directly adjacent to Evans River, a permanently open sea riverine estuary. Riverine estuaries provide an outflow of nutrients, resulting in highly productive nearshore areas and typically an increase in prey species [39,40]. This may explain the higher shark abundances observed at Evans Head, as well as Forster, which is adjacent to the Wallamba River and Wallis Lake. Forster had the highest observations per flight effort and the second-highest abundance overall. Prey availability often influences the distribution, abundance, and behaviour of marine predators, particularly neonate and juvenile sharks [21,41], and although the biomass of prey was not quantified, our results indicated that other marine wildlife, including fish, rays, and turtles, were present on 45.9% of days where shark observations occurred.

The highest abundance of white sharks was reported at Forster, followed by Anna Bay, Crescent Head and Pambula. White shark observation days to total flight days, however, were highest at Hawks Nest. Excluding Pambula, these beaches are within a recognised white shark nursery area on the central coast of NSW, which may explain the higher abundances at these locations [42,43,44].

Bull sharks had the highest abundances recorded at Anna Bay and Ballina, although Anna Bay and Scotts Head had the highest bull shark observation days to total flight days. Bull sharks are widely distributed in tropical and sub-tropical coastal waters, typically undergoing seasonal migrations along the east coast of Australia [45,46,47]. Not all bull sharks, however, exhibit these migration tendencies, with some individuals remaining in the same region for extended periods [25,45,48].

Greynurse sharks are a critically endangered species on the east and west coasts of Australia [49]. Although greynurse sharks can exhibit seasonal site fidelity, they are also known to migrate long distances among aggregation locations [50]. Greynurse observations occurred at only five of the surveyed beaches, with the highest abundance at Forster. The Pinnacle, located just offshore from Forster, is one of 13 recognised greynurse aggregation sites within the NSW marine estate and is considered critical habitat for this species [51,52]. This, along with several nearshore aggregation sites adjacent to Main Beach at Forster, may explain the high abundance of greynurse observations off adjacent beaches. However, aggregation sites are dispersed along the NSW coastline [53], many of which are relatively close to beaches that reported no greynurse sightings.

### 4.2. Factors Affecting Species-Specific Shark Distribution

Coastal estuaries are highly dynamic transitional zones that link together freshwater and marine habitats [54,55]. Many shark species utilise estuarine habitats for foraging, as well as nurseries and pupping sites [50,56,57]. The distance to permanent open sea estuaries was found to be a significant factor for white and greynurse shark occurrence, with white shark observations found to be positively correlated with distance to estuaries. Research has shown that white sharks do not typically enter estuaries or rivers (P Butcher, NSW DPI, unpublished data), but they do have high detection rates at the opening of or within large estuaries [33,58]. In contrast, greynurse sharks were sighted more frequently the closer the beach was to an estuary.

Bull shark abundances have previously been reported to increase in nearshore areas that are within proximity to estuaries [13]. Mature female bull sharks use upper river systems for pupping sites and have been known to travel great distances into riverine estuaries to give birth within freshwater habitats [23,59]. Additionally, bull, dusky whaler (*Carcharhinus obscurus*), bronze whaler (*Carcharhinus brachyurus*), and blacktip (*Carcharhinus limbatus*) sharks inhabit coastal estuaries as neonates and as juveniles, suggesting these estuarine ecosystems play a fundamental role in the early life history stages of multiple whaler species [56,57,60]. However, this study did not find the distance to the nearest estuary to be significant for bull or whaler shark sightings, although this may be due to increased activity by bull sharks and other whaler species at night when drone surveying did not occur [11,60].

Due to the structural complexity of headlands, oceanic islands and adjacent rocky reefs, these habitats are typically associated with enhanced marine life [61,62,63]. Consequently, greater prey abundances may influence the presence of top consumers [64,65]. Greynurse sharks are known to aggregate around rocky reefs, caves, and headlands [19,66]; thus, as to be expected, the chance of observing a greynurse shark increased as the distance to headlands decreased. In contrast to greynurse sharks, distance to headlands did not explain significant variation for sightings of white, bull, or whaler sharks. Additionally, distance to islands was the best predictor for whaler shark occurrence, and the closer to an island a beach was located, the higher the probability of observing greynurse, bull, and whaler sharks. While the trend was the same for white sharks, the distance to islands did not explain significant variation in their frequency of observations.

Water temperature and wave height also significantly influenced shark occurrence. Water temperature is considered to be an important driver for shark distribution within coastal habitats [12,67], and in this study was found to be the best predictor for white and greynurse shark occurrence and a significant factor for white, bull, greynurse and whaler shark observations. White shark observations occurred in temperatures between 16.7 and 25.7 °C, and greynurse sharks were observed in temperatures between 20.5 and 22.0 °C. Bull and whaler shark observations were in similar temperature ranges to each other, from 17.8 to 25.6 °C and 17.8 to 26.2 °C, respectively. Previous research has found ocean temperature to significantly influence white [44,68], greynurse [51] and whaler shark distribution [69]. Greynurse sharks have previously been reported spending ~95% of their time within water temperatures between 17 and 24 °C [51]. Furthermore, white, bull and whaler species all indicated increases in abundance within narrow temperature ranges (i.e., not too hot, or not too cold). This range varied among species, with increases in white shark observations in water temperatures of ~19.5 and ~21.5 °C and of 21.0 to 22.5 °C for bull and whaler sharks. Similarly, increases in bull shark abundances have previously been reported in water temperatures of ~22.0 °C [24], and blacktip shark (*Carcharhinus limbatus*) abundance increased in water temperatures of ~21.5 and ~24.0 °C [70]. It remains unclear, however, whether correlations between shark presence and water temperature reflect temperature-driven variation in prey availability, an increase in shark activity within warmer temperatures or other factors [69,71,72].

Wave height was found to be the best predictor for bull shark occurrence and a significant factor for white, bull, and greynurse sharks, as well as for whaler species. Although observations occurred in different wave height ranges, most observation days occurred within a wave height range of 1.4 to 1.6 m. This could be related to the appropriateness of conditions for sharks to move into the turbulent surf zones behind ocean beach breaks; however, it was also an extremely common wave height for exposed ocean beaches on the NSW coast.

### 4.3. Shark Detectability

Environmental conditions, including water clarity and sea state, can affect shark detectability during drone surveys [30,31,73]. Observations occurred across all classifications of water clarity; however, most observations occurred in good to poor water clarity ranges. Additionally, observations occurred on days recording a Beaufort Sea State of 1 to 3. These results demonstrate that drones are effective for detecting sharks off ocean beaches across a wide variety of environmental conditions; nonetheless, they also indicated that the effectiveness of drone-based monitoring may decrease significantly once environmental conditions (e.g., water clarity) become very poor [32].

Although aerial surveys are effective for sampling shark presence off ocean beaches, there are challenges with using these methods [32,36]. Aspects of our study were limited by the drones’ capabilities, such as flight time, distance surveyed and shark identification in real-time. Guitarfish species, for example, are commonly misidentified as white sharks in real-time during drone surveillance of beaches [34], resulting in multiple beach evacuations for a species that poses no threat to humans. Due to the abundant and wide-ranging distribution of guitarfish along the NSW coastline, this can be problematic for drone monitoring programs. These limitations are, however, being overcome by rapidly improving drone technology and the easing of aviation regulations. For example, developments in deep-learning convoluted neural networks are improving computer-based shark identification in real-time [73,74,75]. Additionally, hybrid petrol-electric multi-rotor drones are routinely exceeding 2-h flight times, and some civil aviation authorities are making beyond-visual-line-of-sight operations (e.g., beyond 2.0 km from the ground control station) more cost-effective [31]. Another limitation of our drone surveys was that we only sampled in the austral spring, summer, and autumn months. This was because the drone monitoring was for shark bite mitigation and, as such, focused on the times when beach attendance was highest [76]. Future studies may benefit from additional surveying in the austral winter months to further understand shark composition off ocean beaches year-round.

## 5. Conclusions

Establishing nearshore habitat use by coastal sharks can support evidence-based management strategies to ensure their conservation and sustainability [12,17]. The present study demonstrates that existing drone-based monitoring to reduce shark bite incidents on the NSW coast in eastern Australia can also provide valuable information about the distribution, abundance, and ecology of coastal sharks. The overarching result from this research was that potentially dangerous sharks, and sharks in general, are rare off NSW beaches. This supports the contention that shark conservation is a growing concern and, as such, this should be first and foremost when establishing management programs for shark fisheries, shark bycatch and shark bite mitigation programs. While the present study has established an extensive baseline of abundance and occurrence of shark assemblages off ocean beaches of NSW, Australia, continuing this program will not only support safer beaches but will also allow the assessment of population trends of various coastal sharks.

## Figures and Tables

**Figure 1 biology-11-01456-f001:**
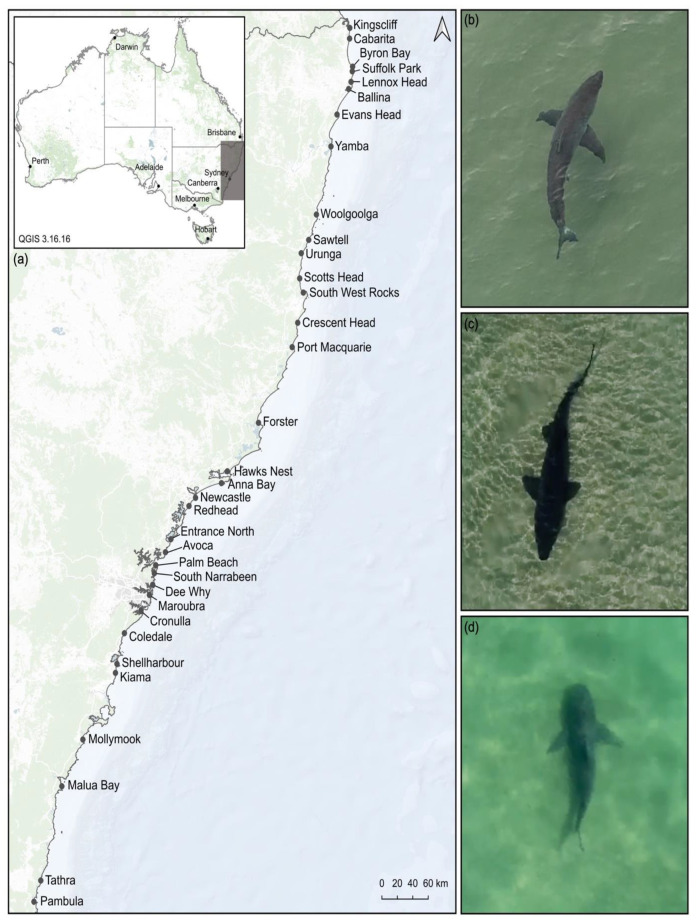
(**a**) Locations of beaches studied using drone surveying techniques along the coastline of New South Wales, Australia. Examples of shark species observed using drones ((**b**) a white shark (*Carcharodon carcharias*); (**c**) a greynurse shark (*Carcharias taurus*); and (**d**) a bull shark (*Carcharhinus leucas*)) during the study.

**Figure 2 biology-11-01456-f002:**
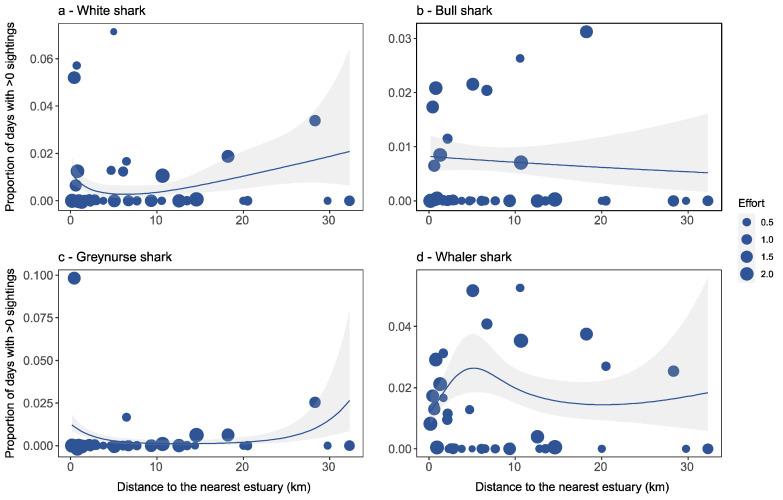
Detection rates for each beach location in relation to distance to the nearest estuary for (**a**) white shark, (**b**) bull shark, (**c**) greynurse shark, and (**d**) whaler shark species, and estimated response curves with a 95% confidence region. Points are sized proportionally to the total flight effort at each location proportional to the average effort. Thus, a weight of 1.0 equals average flight effort, <1 equals below average effort, and >1 equals above average effort.

**Figure 3 biology-11-01456-f003:**
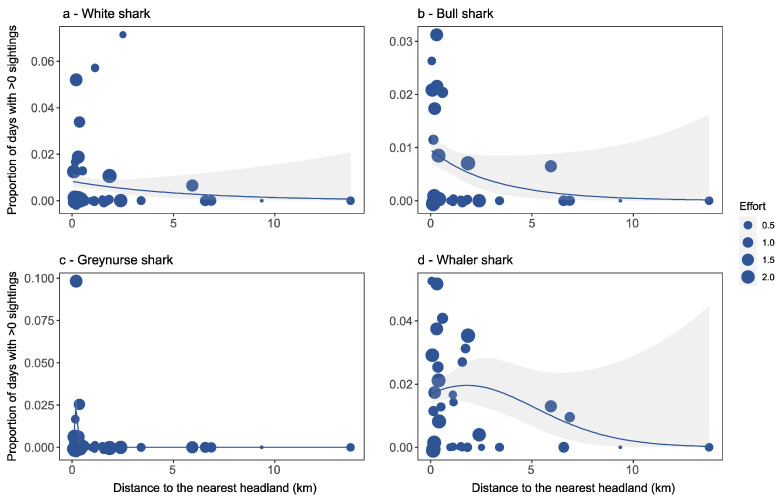
Detection rates for each beach location in relation to distance to the nearest headland for (**a**) white shark, (**b**) bull shark, (**c**) greynurse shark, and (**d**) whaler shark species, and estimated response curves with a 95% confidence region. Points are sized proportionally to the total flight effort at each location proportional to the average effort. Thus, a weight of 1.0 equals average flight effort, <1 equals below average effort, and >1 equals above average effort.

**Figure 4 biology-11-01456-f004:**
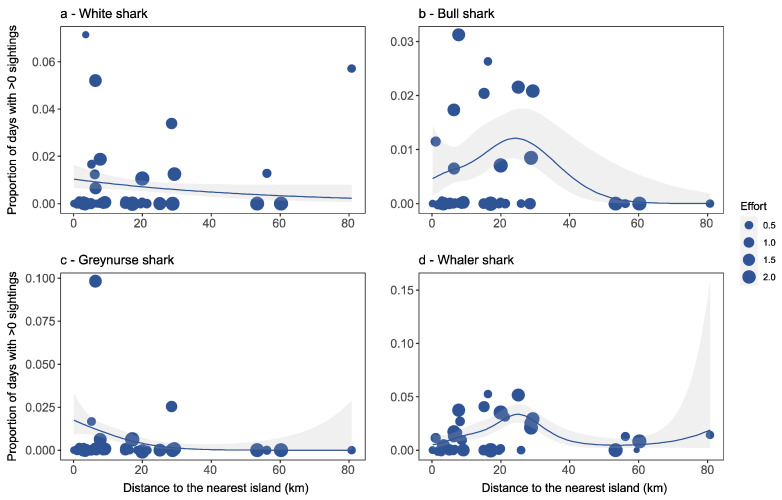
Detection rates for each beach location in relation to distance to the nearest island for (**a**) white shark, (**b**) bull shark, (**c**) greynurse shark, and (**d**) whaler shark species, and estimated response curves with a 95% confidence region. Points are sized proportionally to the total flight effort at each location proportional to the average effort. Thus, a weight of 1.0 equals average flight effort, <1 equals below average effort, and >1 equals above average effort.

**Figure 5 biology-11-01456-f005:**
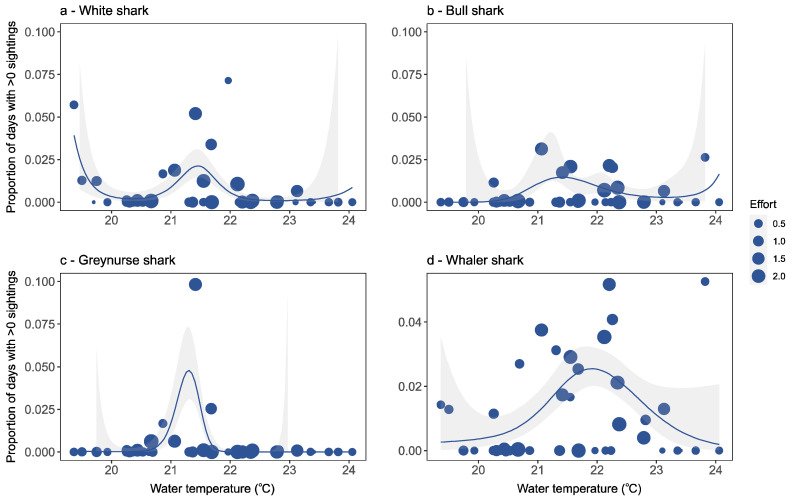
Detection rates for each beach location in relation to water temperature for (**a**) white shark, (**b**) bull shark, (**c**) greynurse shark, and (**d**) whaler shark species, and estimated response curves with a 95% confidence region. Points are sized proportionally to the total flight effort at each location proportional to the average effort. Thus, a weight of 1.0 equals average flight effort, <1 equals below average effort, and >1 equals above average effort.

**Figure 6 biology-11-01456-f006:**
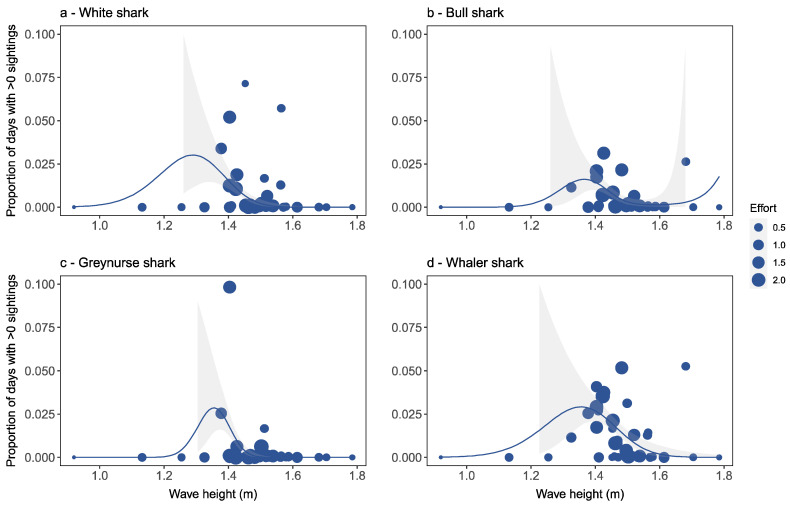
Detection rates for each beach location in relation to wave height for (**a**) white shark, (**b**) bull shark, (**c**) greynurse shark, and (**d**) whaler shark species, and estimated response curves with a 95% confidence region. Points are sized proportionally to the total flight effort at each location proportional to the average effort. Thus, a weight of 1.0 equals average flight effort, <1 equals below average effort, and >1 equals above average effort.

**Table 1 biology-11-01456-t001:** Akaike Information Criteria (AIC) for each model of detection probability in response to intercept only (null), distance (m) to the nearest estuary, headland, and island habitats, as well as water temperature (°C), and wave height (m) for each shark genus. Parentheses enclose significance tests for spline coefficients equal to zero.

Parameter	Null	Estuary	Headland	Island	Water Temperature	Wave Height
White sharks	119.4	117.8 (*)	119.3 (NS)	117.9 (NS)	**86.5** (***)	116.6 (*)
Bull sharks	86.4	87.7 (NS)	83.3 (NS)	80.2 (*)	82.3 (*)	**78.0** (*)
Greynurse sharks	128.4	114.1 (**)	70.7 (***)	80.9 (**)	**68.9** (*)	97.4 (***)
Whaler sharks	138.9	140.9 (NS)	138.8 (NS)	**119.8** (***)	127.4 (**)	134.1 (*)

Note. Entries in bold type indicate the best predictor for shark occurrence for each genus (lowest AIC); Note. *p*-values: *** *p* < 0.001, ** *p* < 0.01, * *p* < 0.05, NS *p* > 0.05.

## Data Availability

Data for this project are maintained by NSW Government, Department of Primary Industries. The data are not publicly available due to confidentiality reasons.

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
