# Peer review of "The Relative Abundance and Occurrence of Sharks off Ocean Beaches of New South Wales, Australia"

_biology, 2022, doi:10.3390/biology11101456_

Round 1

Reviewer 1 Report

The authors make use of an extensive collection of drone footage to better understand correlates of shark occurrence. This was a well justified study and I really enjoyed reading it!  This research demonstrates the utility of drones for nearshore monitoring. It also contributes to our understanding of shark occurrence, providing insights into dangerous sharks as well as critically endangered species. Overall, this was an impressive study with several strengths – the large number of drone flights, the work extended over 4 years and sampling occurred over more than 8 degrees of latitude. This novel work will be of interest to the readership of Biology.

The challenge that this work faces is that the abiotic drivers the authors are exploring will likely alter the detectability of shark species, but their wave height data indicates that this is not the case. My only real concern with this work rests on the effect of outliers on their data analysis.  Specifically, I was concerned that relationships with greynurse sharks may be strongly influenced by an outlying site – Forster.  It would be very useful for the authors to report outcomes in the absence of this outlier (along with it) as I suspect that some of their significant outcomes hinge on this single point.

The size of animals was one variable that was not explored.  They make mention of the presence of juveniles of several carcharinid species associated with estuaries. I appreciate that there may not be sufficient data to make such analyses meaningful, but perhaps something to consider in future contributions.

Finally, the authors make mention of real time shark identification on line 392 -  For an example of real time shark identification in ‘the cloud’ following by real time alerting to swimmers using wearable smart technologies the authors may want to cite:

Gorkin R., K. Adams, M. Berryman, S. Aubin, W. Li, A.R. Davis & J. Barthelemy (2020) Sharkeye: Real-Time Autonomous Personal Shark Alerting via Aerial Surveillance. Drones 4: 18. https://doi.org/10.3390/drones4020018

Minor distractions:

Line 11 & 61 – change variations to variation 

Line – 31 – clumsy wording “Whaler sharks (Carcharhinus spp.) recorded the highest number of observations” – researchers recorded, not the sharks

Line 141 – “each shark taxa” – taxa is plural, change to taxo

Table 1 caption – needs to specific what the asterisks mean – for most researchers this will be self-evident, but this should be done for completeness

Reviewer 2 Report

This study uses a remarkably large dataset of drone flights yielding shark observations along NSW, Australia beaches. I have concerns with the analytical approach as model testing solely looked at individual covariates and did not attempt to build models with multiple covariates nor interactions thereof. This precludes a rigorous assessment of whether multiple factors contribute to shark occurrence patterns, as well as the relative importance of those factors. In addition, model validation is not discussed in the methods nor results, precluding knowledge of whether statistical assumptions have been appropriately met in generating these models (and, thus, if the results are robust). Other metrics of model value and accuracy are not provided, such as deviance explained or AUC. There is clearly value in the dataset and work done here, I just believe that it requires substantial analytical revision.

Line 122-124: Perhaps give an example of prey being present, such as obvious schools of forage fish. Presumably, there are always some prey species around, just not necessarily at an abundance easily detectable from the drone. Similarly, the sighted shark species forage on functionally different prey groups (e.g., greynurse vs white shark), so it will need to be made clear how this was accounted for when stating whether that species’ prey was present or not.

Line 147-150: Some explanation of the decision to weight data points by flight time rather than include an offset or spline for flight time (thereby accounting for sampling effort) is needed. The latter is a seemingly more common approach, especially in the context of fisheries.

Lines 151-155: Testing individual splines is an atypical approach, as most model comparisons stem from a forward or backward selection process where multiple covariates (and their interactions) may end up included in a given species’ model. This is an important consideration, given that inclusion of the most significant covariate may explain variance to a point where other covariates do not further improve the model (and so would be considered unimportant). What this framework currently lacks is an appropriate assessment of the relative importance of these different covariates. For example, from table 1 it seems that, for white sharks, water temperature and wave height are likely to be the most important covariates but there is no way of distinguishing which is more important (or if they both truly are) since the analysis does not attempt to include both in a single model and compare that against a simpler model (such as via AIC, AIC weight, etc).

Line 276-278: This sentence is a perfect example of the above point; at no point were these all tested together in a single model of a species’ occurrence. Also, the use of the word “abundance” seems incorrect given that your response is a proportion (based on presence/absence) and, thus, you are measuring “occurrence”.
